Characterization of Staphylococcus epidermidis clinical isolates from hospitalized patients with bloodstream infection obtained in two time periods

http://orcid.org/0000-0002-0157-5696 Martínez-Santos Verónica I. 1
Torres-Añorve David A. 2
Echániz-Aviles Gabriela 3
Parra-Rojas Isela 4
http://orcid.org/0000-0002-7037-6412 Ramírez-Peralta Arturo 5
Castro-Alarcón Natividad 2 natividadcastro24@gmail.com
1 CONACyT, Universidad Autónoma de Guerrero , Chilpancingo, Guerrero , Mexico
2 Laboratorio de Investigación en Microbiología, Facultad de Ciencias Químico Biológicas, Universidad Autónoma de Guerrero , Chilpancingo, Guerrero , Mexico
3 Centro de Investigación Sobre Enfermedades Infecciosas, Instituto Nacional de Salud Pública , Cuernavaca, Morelos , México
4 Labotatorio de Investigación en Obesidad y Diabetes, Facultad de Ciencias Químico Biológicas, Universidad Autónoma de Guerrero , Chilpancingo, Guerrero , Mexico
5 Laboratorio de Investigación en Patometabolismo Microbiano, Facultad de Ciencias Químico Biológicas, Universidad Autónoma de Guerrero , Chilpancingo, Guerrero , Mexico
Nicolás Marisa
Electronic publication date: 2022 Oct 4
Publication date: 2022
Volume: 10
Electronic Location ID: e14030
Received 2022 Apr 11; Accepted 2022 Aug 16
Copyright: © 2022 Martínez-Santos et al.
Copyright year: 2022
Copyright holder: Martínez-Santos et al.
License: This is an open access article distributed under the terms of the Creative Commons Attribution License, which permits unrestricted use, distribution, reproduction and adaptation in any medium and for any purpose provided that it is properly attributed. For attribution, the original author(s), title, publication source (PeerJ) and either DOI or URL of the article must be cited.
License URL: https://creativecommons.org/licenses/by/4.0/

Keywords: S. epidermidis, Health care-associated infections, Sequence type, Bacteremia

Funding: The authors received no funding for this work.

==============================
Background

In recent years Staphylococcus epidermidis has been considered an important and frequent causative agent of health care-associated infections (HAIs), increasing the costs of hospitalization, morbidity, and mortality. Antibiotic resistance and biofilm formation are the most important obstacles in the treatment of infections caused by this microorganism. The aim of this work was to determine the most prevalent STs, as well as the antibiotic resistance profile and biofilm formation of S. epidermidis clinical isolates obtained from hospitalized patients in two hospitals in Acapulco, Guerrero in two time periods.

Methods

Twenty methicillin-resistant S. epidermidis strains isolated from patients with bacteremia in two hospitals in two time periods were analyzed. Identification and antibiotic susceptibility were performed using the Vitek automated system. Molecular confirmation of the identification and methicillin resistance was performed by duplex PCR of the mecA and nuc genes. Biofilm production was analyzed, and the clonal origin was determined by multilocus sequence typing (MLST).

Results

We identified 14 antibiotic resistance profiles as well as 13 sequence types (ST), including the new ST761. We also found that ST2 and ST23 were the most prevalent and, together with ST59, were found in both time periods. Seventeen of our clinical isolates were multidrug-resistant, but all of them were sensitive to linezolid and vancomycin, and this was not related to biofilm production. Additionally, we standardized a duplex PCR to identify methicillin-resistant S. epidermidis strains. In conclusion, S. epidermidis STs 2, 23, and 59 were found in both time periods. This study is the first report of S. epidermidis ST761. The clinical isolates obtained in this work showed a high multidrug resistance that is apparently not related to biofilm production.

Introduction

Staphylococcus epidermidis is a gram-positive, facultative anaerobic bacterium that is present in the skin and mucus membranes of humans and other mammals as part of the microbiota and is usually considered a commensal organism (Huttenhower et al., 2012; Paharik & Horswill, 2016). However, nowadays it is seen as an opportunistic pathogen, since it is the most common coagulase-negative staphylococci (CoNS) responsible of health care-associated infections (HAIs) (Otto, 2009). HAIs, formerly known as “nosocomial” or “hospital” infections, are defined as infections acquired by patients while receiving medical care in health-care facilities. These infections are one of the main causes of morbidity and mortality among hospitalized patients (Hsu, 2014). The World Health Organization (WHO) estimates that seven out of 100 hospitalized patients in developed countries, or 10 out of 100 in developing ones, will acquire at least one HAI at any given time (WHO, 2020). The most important HAIs, according to frequency and severity, are those related to medical procedures and devices, like surgical site infections and catheter-associated infections. Around 50–70% of all cases of HAIs are associated with medical devices (Bryers, 2008; Guggenbichler et al., 2011).

S. epidermidis has been reported to be the main cause of central line-associated bloodstream infections, the second-most-common cause of surgical site infections, and the third-most-common cause of all HAI, becoming an important human pathogen that affects preferentially immunocompromised, long-term hospitalized and seriously sick patients (Schoenfelder et al., 2010; Sievert et al., 2013). Even though infections caused by S. epidermidis are usually non-life-threatening, probably because of the lack of classic virulence factors, the danger resides in the fact that they are extremely difficult to treat due to biofilm formation and antibiotic resistance (Otto, 2009).

Biofilm production is the main virulence factor of S. epidermidis; in particular, its ability to form it on indwelling medical devices is critically important in nosocomial bacteremia and infection of prosthetic medical devices. The biofilm is a mixture of several adhesive molecules, including polysaccharide intercellular adhesin (PIA), proteinaceous factors (Bhp, Aap, and Embp), teichoic acids, and extracellular DNA; however, it has been shown that not all isolates encode factors involved in biofilm formation. For example, the operon responsible for synthesizing PIA (icaADBC) is present in most clinical isolates, but some clinically relevant isolates are PIA negative (Fey & Olson, 2010; Foster, 2020). Biofilm keeps bacteria bound to inert surfaces, allows them to escape the host’s immune system, and protects them against antibiotics (Gotz, 2002). Besides the protection against antibiotics conferred by the biofilm, S. epidermidis has acquired genes that confer resistance to several antibiotics, including rifampicin, gentamicin, fluoroquinolones, and erythromycin, among others. The most important antibiotic resistance is methicillin resistance conferred by the mecA gene, which encodes a penicillin-binding protein (PBP2a) that has a low affinity for beta-lactam antibiotics (Hiramatsu et al., 2001) and is contained within the Staphylococcal Cassette Chromosome mec (SCCmec) (IWG-SCC, 2009; Chambers, 1997). It has been reported that approximately 80% of the isolates from device-associated infections are resistant to methicillin, which is the antibiotic of first choice against these infections (Diekema et al., 2001; Kozitskaya et al., 2004). In addition, healthcare associated strains have high antibiotic resistance, being resistant to most of the antibiotics commonly used (Chabi & Momtaz, 2019).

The genome of S. epidermidis is prone to horizontal gene transfer of mobile genetic elements and to chromosomal recombination, which gives rise to the emergence of different clones (Miragaia et al., 2008; Miragaia et al., 2007). PFGE has been considered the gold standard for S. epidermidis molecular typing since it is a clinically valuable tool for evaluating short time epidemiological infections and is still the most discriminating method. However, several studies that used PFGE for the classification of nosocomial S. epidermidis isolates have shown that there is an extensive diversity among populations. Due to this, the usefulness of this method to evaluate S. epidermidis in long-term global epidemiology has been debated (Widerstrom et al., 2009). To aid in this subject, the genetic relatedness between different isolates is analyzed by MLST, a technique based on sequencing of conserved housekeeping genes. This technique allows the comparison of isolates, even from different countries as well as the naming of clones from all over the world. Nowadays, a widely recognized MLST scheme and database have been developed for S. epidermidis (Thomas et al., 2007).

It has been shown that S. epidermidis clones that persist and spread within the hospital setting, can arise in the community or be endemic in the hospital (Brito et al., 2009; Widerstrom et al., 2006). Previously, we showed that S. epidermidis clinical isolates obtained in the General Hospital of Acapulco from patients diagnosed with nosocomial infections, formed two clusters that were able to persist in different nosocomial areas at least for 3 years (Castro-Alarcon et al., 2011). The aim of this work was to identify the ST of methicillin-resistant S. epidermidis (MRSE) clinical isolates from two hospitals in Acapulco, Guerrero, Mexico, isolated in two different periods, as well as to determine their antibiotic resistance profile and biofilm production.

Materials and Methods

Study subjects

Clinical isolates were obtained from patients hospitalized in different areas (neonatology (6/20), pediatrics (5/20), emergency room (2/20), internal medicine (4/20), intensive care unit (2/20), and gynecology (1/20)) of two tertiary care hospitals in Acapulco, Gro., Mexico in two periods: Acapulco General Hospital, between 2003 and 2004, and Vicente Guerrero Hospital in 2017. All patients were diagnosed with bacteremia and the clinical isolates were considered the etiological agents of the infection by the Epidemiological Surveillance Committee of each hospital. One S. epidermidis clinical isolate was obtained from each patient. All the isolates were obtained from blood cultures. The study was approved by the Research Ethics Committee of the Autonomous University of Guerrero (CB-001/22).

Clinical isolates and susceptibility testing

A total of 20 S. epidermidis clinical isolates were analyzed. Ten strains were isolated between 2003 and 2004, and 10 were isolated in 2017. Isolates were grown overnight in Muller Hinton agar (BD, Heidelberg, Germany) at 35 °C and stored at −80 °C in 15% (vol/vol) glycerol and LB broth (Dibico, Cuautitlán Izcalli, Mexico). Strain identification, as well as antimicrobial susceptibility testing, were performed using the Vitek®2 automated system (bioMérieux, Marcy-l’Étoile, France), which uses a fluorogenic methodology for organism identification, as well as a turbidimetric method for susceptibility testing that allows it to indicate the MIC for each antibiotic tested (Table S1). Antibiotics assayed were: benzylpenicillin, gentamicin, levofloxacin, erythromycin, quinupristin/dalfopristin, rifampicin, trimethoprim/sulfamethoxazole, clindamycin, cefoxitin, oxacillin, linezolid, and vancomycin. Antibiotic resistance or susceptibility classification was performed according to the breakpoints established by the CLSI (Clinical & Laboratory Standards Institute (CLSI), 2017). Methicillin resistance was determined by cefoxitin and oxacillin testing (Clinical & Laboratory Standards Institute (CLSI), 2012).

Molecular confirmation

Molecular confirmation of S. epidermidis identification and methicillin resistance was performed by duplex PCR, using specific primers for the nuc and mecA genes, respectively (Table 1). The nuc gene encodes a conserved thermonuclease present in all staphylococci species, except those of the S. sciuri group, and shows moderate sequence diversity, which allows for species identification with a 100% sensitivity and specificity using specific primers for each species (Hirotaki et al., 2011).

Table 1 Oligonucleotides used in this work.

Oligonucleotide	Sequence (5′–3′)	Amplicon size (bp)	Amplicon used for allele assignment (bp)	Reference	
epi-F (nuc)	TTGTAAACCATTCTGGACCG	251	NA	(Hirotaki et al., 2011)	
epi-R (nuc)	ATGCGTGAGATACTTCTTCG	
mecA-F	TGGCTATCGTGTCACAATCG	310	NA	(Vannuffel et al., 1995)	
mecA-R	CTGGAACTTGTTGAGCAGAG	
arcC-F	TGTGATGAGCACGCTACCGTTAG	508	465	(Thomas et al., 2007)	
arcC-R	TCCAAGTAAACCCATCGGTCTG	
aroE-F	CATTGGATTACCTCTTTGTTCAGC	459	420	
aroE-R	CAAGCGAAATCTGTTGGGG	
gtr-F	CAGCCAATTCTTTTATGACTTTT	508	438	
gtr-R	GTGATTAAAGGTATTGATTTGAAT	
mutS-F3	GATATAAGAATAAGGGTTGTGAA	608	412	
mutS-R3	GTAATCGTCTCAGTTATCATGTT	
pyr-F2	GTTACTAATACTTTTGCTGTGTTT	851	428	
pyr-R4	GTAGAATGTAAAGAGACTAAAATGAA	
tpi-F2	ATCCAATTAGACGCTTTAGTAAC	592	424	
tpi-R2	TTAATGATGCGCCACCTACA	
yqiL-F2	CACGCATAGTATTAGCTGAAG	658	416	
yqiL-R2	CTAATGCCTTCATCTTGAGAAATAA	
Note:

NA, not applicable.

To obtain total DNA, each isolate was grown overnight in BHI broth (BD, San Jose, CA, USA) at 35 °C with shaking. Then 1 ml of each culture was centrifuged at 13,000 rpm for 12 min, the pellets were washed with 200 μl of sterile water and resuspended in 100 μl of water. The samples were frozen at −20 °C for 30 min, boiled for 5 min, frozen again for 5 min, and centrifuged at 13,500 rpm for 10 min. The supernatant was recovered, and the DNA was precipitated with 250 μl of cold isopropanol. The DNA pellets were washed with 250 μl of 70% ethanol, dried, and resuspended in 100 μl of water.

Each PCR was performed in a final volume of 50 μl, containing buffer 1X, 1.5 mM MgCl2, 200 μM dNTPs, 0.6 μM of each oligonucleotide, 120 ng DNA, and 1 U Taq DNA polymerase (Roche, Indianapolis, IN, USA). The conditions used were: 1 cycle at 95 °C for 5 min; 30 cycles at 95 °C for 1 min, 56 °C for 1 min, and 72 °C for 30 s; and 1 cycle at 72 °C for 10 min. Amplicons were analyzed by 2% agarose gel electrophoresis. S. epidermidis ATCC 35984 and S. aureus ATCC 29213 were used as positive and negative controls, respectively.

Multilocus sequence typing

All strains were analyzed by the MLST protocol described by Thomas et al. (2007) using primers listed in Table 1. PCRs were performed in a 50 μl final volume, containing 1 U of Phusion HF DNA polymerase (Thermo Fisher Scientific, Waltham, MA, USA), 1X HF buffer, 200 μM dNTPs, 0.24 μM of each oligonucleotide, and 120 ng of DNA. Conditions used were 1 cycle at 95 °C for 3 min; 34 cycles at 95 °C for 30 s, 50 °C for 1 min, 72 °C for 1 min; and 1 cycle at 72 °C for 10 min. The amplified products were further purified using the PureLink PCR purification kit (ThermoFisher Scientific, Waltham, MA, USA) following the manufacturer’s instructions and sequenced in an ABI PRISM 310 Genetic Analyzer (Applied Biosystems, Waltham, MA, USA) using the BigDye Terminator kit v.1.1 (Applied Biosystems, Waltham, MA, USA) and primers listed in Table 1. The sequences were analyzed using the SeqMan Pro 14.1 software (DNASTAR). The allelic profiles, consisting of the allele number for each of the seven genes, were assigned to sequence types (STs) using the Staphylococcus epidermidis MLST website (https://pubmlst.org/sepidermidis/, University of Oxford). S. epidermidis ATCC 12228 was used as a control.

Use of the goeBurst algorithm

The clonal relationship of the isolates was analyzed using the goeBURST algorithm (http://www.phyloviz.net/), which uses multilocus sequence typing data and clusters the STs according to their number of single, double, and triple locus variants, and occurrence frequency, in order to identify alternative patterns of descent based on the number of differences between numeric profiles (Francisco et al., 2009). The profile data and the auxiliary data files containing the ST and the alleles of the genes of each clinical isolate used in our work or registered in the S. epidermidis database (https://pubmlst.org/sepidermidis), respectively, were uploaded as profile data on the online tool. The program was run by hitting the “launch tree” button, and the tree was constructed using the default settings.

Biofilm quantification

Biofilm quantification was performed according to Stepanovic et al. (2007). Briefly, the isolates were grown overnight in Muller Hinton agar at 35 °C. Then, bacteria were collected in sterile distilled water adjusting to an OD600 = 0.1. Twenty μl of each culture were used to inoculate 180 μl of TBS broth supplemented with 1% glucose in 96-wells plates, and incubated at 35 °C for 24 h. The bacteria were washed three times with 300 μl of PBS 1X, fixed with 150 μl of methanol, dried overnight, stained with crystal violet 2%, and washed two times. Then the biofilm was resuspended with 95% ethanol and quantified at 570 nm in a Multiskan GO Microplate spectrophotometer (ThermoScientific, Vantaa, Finland). Quantification was done three times by triplicate. The strains were considered as non-producers when OD ≤ 0.3, weak producers when 0.3 < OD ≤ 1.0, and strong producers when OD > 1.0, as per recommendations by Stepanovic et al. (2007).

Results

Phenotypic and molecular identification of S. epidermidis and resistance profile

A total of 20 S. epidermidis clinical isolates identified by the Vitek system were included in this study. The antibiotic resistance testing determined that all the isolates (100%) were resistant to benzylpenicillin and oxacillin, 90% (18/20) were resistant to erythromycin, 80% (16/20) were resistant to trimethoprim/sulfamethoxazole, and 70% (14/20) to clindamycin. None of the clinical isolates was resistant to linezolid and vancomycin. Casually, the proportion of resistant isolates was the same for all antibiotics in both groups of isolates, except for rifampicin, for which we found five resistant isolates in 2003/2004 but only one in 2017. The isolates were then confirmed as methicillin-resistant S. epidermidis by a duplex PCR detecting mecA and nuc genes (Fig. 1).

Figure 1 Representative gel of duplex PCR results.

Fragments corresponding to the mecA and nuc genes are shown. Lanes: MW, molecular weight marker (bp); PC, positive control (S. epidermidis ATCC 35984); NC, negative control (S. aureus ATCC 29213).

We then determined if our clinical isolates were multidrug resistant. We found 14 resistance profiles (Table 2), which include 8 to 3 of the antibiotics tested. Of all the clinical isolates, 85% (17/20) were multidrug-resistant (≥3 antibiotics) (Magiorakos et al., 2012), of these, 20% (4/20) were resistant to eight antibiotics, and 25% (5/20) were resistant to seven antibiotics.

Table 2 Characteristics and resistance profiles of MRSE clinical isolates used in this work.

Resistance profile	Antibiotics	Clinical isolate	ST	Biofilm production	Hospital area	Hospital	Year of isolation	
1	BP, OX, EM, CM, LE, TS, RI, GM	1,069	23	Strong	Neonatology	AGH	2004	
1,091	2	Weak	Pediatrics	AGH	2004	
4,204	23	Weak	ER	VGH	2017	
2	BP, OX, EM, CM, LE, TS, RI, Q/D	1,154	135	Non	Neonatology	AGH	2004	
3	BP, OX, EM, CM, LE, TS, RI	1,047	23	Non	Neonatology	AGH	2004	
1,103	23	Non	Pediatrics	AGH	2004	
4	BP, OX, EM, CM, LE, TS, GM	4,205	2	Non	IM	VGH	2017	
4,208	2	Non	IM	VGH	2017	
4,206	2	Non	ER	VGH	2017	
5	BP, OX, EM, CM, TS, GM	1,126	182	Non	Neonatology	AGH	2004	
4,211	761	Non	IM	VGH	2017	
6	BP, OX, EM, CM, LE, TS	4,212	5	Weak	ICU	VGH	2017	
7	BP, OX, EM, TS, GM	1,032	89	Non	Neonatology	AGH	2003	
8	BP, OX, EM, CM, TS	4,202	59	Weak	Pediatrics	VGH	2017	
9	BP, OX, EM, CM	1,042	59	Weak	Neonatology	AGH	2004	
10	BP, OX, EM, TS	1,161	173	Weak	Gynecology	AGH	2004	
11	BP, OX, EM, GM	4,214	259	Weak	Pediatrics	VGH	2017	
12	BP, OX, GM	585	57	Weak	Pediatrics	AGH	2003	
13	BP, OX, EM	4,201	640	Non	IM	VGH	2017	
14	BP, OX, TS	4,210	193	Non	ICU	VGH	2017	
Note:

BP, benzylpenicillin; OX, oxacillin; EM, erythromycin; CM, clindamycin; Q/D, quinupristin/dalfopristin; LE, levofloxacin; GM, gentamicin; TS, trimethoprim/sulfamethoxazole; RI, rifampicin; ER, emergency room; IM, internal medicine; ICU, intensive care unit; AGH, Acapulco General Hospital; VGH, Vicente Guerrero Hospital.

Identification of S. epidermidis clinical isolates ST

Our molecular analysis identified 13 STs, indicating a high genotypic diversity. Ten of these STs correspond to single isolates, while two correspond to four isolates and one to two isolates (Table 3). The most represented STs were ST2 and ST23, which include four strains each, and together with ST640 have been found in Mexico. One of the STs identified was new, not reported previously. The information of this new ST was sent to the curator of the MLST database and was assigned the number 761. ST761 (arcC 28, aroE 3, gtr 5, mutS 5, pyrR 11, tpiA 4, yqiL 4) (Table 3) isolated in 2017 is resistant to benzylpenicillin, oxacillin, erythromycin, clindamycin, trimethoprim/sulfamethoxazole, gentamicin, and non-biofilm producer (Table 2).

Table 3 MRSE sequence types identified in this study and countries where they have been found.

ST	MLST profile
(arc, aroE, gtr, mutS, pyrR, tpi, yqiL)	Clinical isolates	Countries (number of isolates)a	
2	7, 1, 2, 2, 4, 1, 1	1,091, 4,205, 4,206, 4,208	Argentina (2), Brazil (18), Bulgaria (1), Cape Verde (2), Colombia (1), Denmark (2), France (1), Germany (9), Greece (2), Hungary (2), Iceland (2), Iran (5), Ireland (1), Italy (2), Japan (3), Mexico (2), Netherlands (1), Poland (2), Russia (1), South Africa (4), South Korea (19), Spain (2), UN (2), Uruguay (2), USA (12)	
5	1, 1, 1, 2, 2, 1, 1	4,212	Bulgaria (1), Cape Verde (1), Denmark (2), Germany (10), Iceland (1), Iran (3), Ireland (1), Norway (1), Poland (2), Russia (2), South Korea (2), UN (2), USA (5)	
23	7, 1, 2, 1, 3, 3, 1	1,047, 1,069, 1,103, 4,204	Argentina (1), Brazil (5), Germany (2), Greece (1), Hungary (1), Iceland (1), Mexico (1), Poland (1), Portugal (1), Uruguay (1), Russia (4), USA (1)	
57	1, 1, 1, 1, 2, 1, 1	585	Germany (1), India (2), Portugal (1)	
59	2, 1, 1, 1, 2, 1, 1	1,042, 4,202	Brazil (8), Germany (5), Greece (1), India (1), Ireland (1), Russia (21), South Africa (1), South Korea (4)	
89	1, 1, 2, 1, 2, 1, 1	1,032	Brazil (1), China (2), Denmark (1), Iceland (1), Netherlands (1), Norway (1), Portugal (2), USA (2)	
135	7, 1, 2, 1, 3, 1, 1	1,154	UN (1)	
173	1, 6, 6, 1, 2, 1, 10	1,161	Germany (1), Poland (1), South Korea (1), USA (1)	
182	12, 1, 5, 2, 3, 1, 20	1,126	USA (1)	
193	12, 1, 9, 8, 6, 5, 8	4,210	Brazil (1), Cambodia (6), South Korea (1)	
259	12, 1, 2, 1, 2, 1, 1	4,214	China (6)	
640	28, 3, 13, 5, 8, 9, 11	4,201	Mexico (1), Spain (1)	
761b	28, 3, 5, 5, 11, 4, 4	4,211	NA	
8	2, 1, 7, 1, 1, 1, 1	ATCC 12228 control	Canada (1), USA (1)	
Notes:

UN, Unknown; NA, Not applicable.

a Countries and number of isolates according to the MLST database: https://pubmlst.org/sepidermidis/ (until February 2022).

b New ST found in this work.

STs 23, 2, and 59 were isolated in both periods of time, while the rest were isolated in either period (Fig. 2).

Figure 2 Distribution of clinical isolates STs in time.

Venn diagram of the identified STs by year.

Clonal relationships

In order to establish the clonal relationships of our isolates, we used the algorithm goeBURST. As seen in Fig. 3, STs 2 and 23, previously found in Mexico, are related to ST135, differing in two and one locus variant, respectively. ST640, the other ST previously found in Mexico, is distantly related to them. Interestingly, the new ST761 is more related to ST640, although they differ in four locus variants. STs 5 and 59 are related to ST57, differing only in one locus variant, as well as STs 89 and 259.

Figure 3 goeBURST analysis of MLST data of S. epidermidis clinical isolates obtained in 2003/2004 and 2017.

Each gray circle represents a ST. Each ST is related to the others by the number of allelic differences indicated near the lines.

Biofilm production

As mentioned before, biofilm production is important for the establishment of the infection. We quantified the biofilm production of our isolates. Our results show that eight isolates were weak producers, and 11 were non-producers (Table 2), curiously only one isolate was a strong producer. Of the weak producers, one isolate belongs to ST2 and two belong to ST23, the most frequent STs (Table 2). Both strains of ST59 were also weak producers, while the strain of the new ST identified was a non-producer. The statistical analysis of the OD of the biofilm production showed statistically significant differences among the three categories (p < 0.001) (Fig. 4).

Figure 4 Biofilm production.

Graph of the optical densities (OD570) obtained with S. epidermidis clinical isolates. The bars represent the means of the ODs obtained for each category, and the error bars represent the standard deviation. Strains ATCC 35984 and ATCC 12228 were used as positive (strong producer) and negative (non-producer) controls, respectively. The p-value was calculated using t Student, and p < 0.05 was considered statistically significant.

Discussion

Nowadays, S. epidermidis is a common HAI etiological agent that causes mainly catheter-related bloodstream infections. In the present study, we used MRSE clinical isolates identified by the microbial identification system Vitek2 (bioMérieux, Marcy-l’Étoile, France); however, the identification rate of this system is 93.7% for clinical isolates (Jin et al., 2011). In order to corroborate the identification, we innovated a duplex PCR to amplify the nuc and mecA genes, using oligonucleotides previously published (Table 1). This PCR proved to be a fast and easy way to identify MRSE clinical isolates and has been introduced as the method of choice for S. epidermidis identification at the participant hospitals.

The clinical isolates were obtained from blood cultures of hospitalized patients diagnosed with bacteremia in the city of Acapulco, Guerrero. These isolates were obtained 13 years apart in two different hospitals. We identified 13 STs, five corresponding to isolates obtained in 2003/2004 at the Acapulco General Hospital, represented by one strain each, and five corresponding to strains isolated in 2017 at the Vicente Guerrero Hospital also represented by one strain each. Of the strains obtained in 2017, ST761 has not been previously reported, making this the first report of this new ST. Interestingly, ST761 was isolated in 2017, which could be due to it being an emerging ST or just not previously identified. The other three STs correspond to strains isolated in both time periods and are the most represented: ST2 and ST23, with four strains each, and ST59, with two strains. These STs were found in both hospitals but their distribution over time was different, three strains of ST23 were isolated in 2004 and one in 2017, while the opposite was found for ST2. On the other hand, one strain of ST59 was isolated in 2004 and one in 2017. Our results show that STs 2, 23, and 59 are the most prevalent and persistent in hospital settings in Acapulco, Gro., Mexico. This is consistent with previous reports that have found ST2 to be the most predominant ST in hospital settings in China (Du et al., 2013; Li et al., 2009), Brazil (Iorio et al., 2012), Portugal (Miragaia et al., 2007), USA (Sharma et al., 2014), Germany (Kozitskaya et al., 2004), and Australia (Widerstrom et al., 2012). Specifically, our data are similar to those obtained in Brazil by Iorio et al. (2012), who found that ST2 and ST23 were the most frequent in clinical isolates from hospitals in Rio de Janeiro. S. epidermidis STs 2 and 23 have been reported previously in Mexico, along with STs 640, 46, 61, 71, 7, and 82 (Table 3) (Martinez-Melendez et al., 2016; Miragaia et al., 2007). With this work we add STs 5, 57, 59, 89, 135, 173, 182, 193, 259, 640, and the new 761.

S. epidermidis infections are usually associated with high antimicrobial resistance. When we analyzed antibiotic resistance, we found 14 different resistance profiles. Three isolates were resistant to two different antibiotics, while the rest were resistant to three or more. Besides all isolates being penicillin-resistant, over 70% of our isolates were resistant to erythromycin, clindamycin, and trimethoprim/sulfamethoxazole; between 55% and 30% were resistant to gentamicin, levofloxacin, and rifampicin; and only 5% (1/20) were resistant to quinupristin/dalfopristin. This antibiotic is a streptogramin that was approved by the US-FDA in 1998 as a treatment option for vancomycin-resistant enterococci (Shariati et al., 2020). These results show that clinical MRSE strains have a high antibiotic resistance, which could complicate their treatment. On the bright side, all our clinical isolates were sensitive to linezolid, a synthetic oxazolidinone recommended to treat multidrug-resistant gram-positive infections (Hashemian, Farhadi & Ganjparvar, 2018). Our result is consistent with those of a systematic review and meta-analysis, which reports that linezolid resistance is very low (0.3%) (Shariati et al., 2020). However, our result differs from a study in Mexico in which the authors report a frequency of linezolid resistance of 3.6% (3/83) in S. epidermidis clinical isolates from patients hospitalized in Guadalajara and Monterrey, and that the three isolates belonged to ST23 (Martinez-Melendez et al., 2016). In accordance with this, Campanile et al. (2013) reported that ST23 is more capable of accepting the cfr gene, which mediates linezolid resistance and was first identified in plasmid pSCFS1 (Schwarz, Werckenthin & Kehrenberg, 2000). On the other hand, whole-genome sequencing (WGS) of 176 S. epidermidis clinical isolates from American patients showed that S. epidermidis ST5 and ST22 are related to linezolid resistance, and the authors reported a frequency of 22% (Li et al., 2018). In these isolates, the cfr gene is contained in plasmid pMB151a, which is present in all S. epidermidis ST5 linezolid-resistant clinical isolates (Li et al., 2018).

Our clinical isolates were also sensitive to vancomycin, which is an antibiotic used as a first line drug to treat methicillin-resistant CoNS (Shariati et al., 2020). This result is similar to that reported by Martinez-Melendez et al. (2016), who found 0% (0/83) resistance to vancomycin in clinical isolates of Mexican hospitalized patients. Therefore, vancomycin activity against MRSE clinical isolates suggests that this antibiotic may be suitable in the combat of S. epidermidis associated with HAIs in Mexico. Interestingly, we found no relationship between STs and antibiotic resistance profile. For example, ST23 strains isolated in 2004 have resistance profiles 1 and 3, while the isolate from 2017 has profile 1. ST2 strains isolated in 2017 have profile 4, while the isolate from 2004 has profile 1. This suggests that even though STs 23, 2, and 59 seem to be persistent in Acapulco, Guerrero, there are several circulating strains.

Numerous studies report the ability to form biofilms as a typical feature of many nosocomial isolates (Gotz, 2002); however, we found that almost half of our isolates were weak producers and one strong producer, while the rest were non-producers. Four of the eight weak producers correspond to STs 23, 2, and 59, which were the most frequent, and the strong producer belongs to ST23. The biofilm-producing clinical isolates that correspond to STs 23 and 2 showed a higher antibiotic resistance, since they were resistant to eight antibiotics, while ST59 clinical isolates were resistant to four and five antibiotics. Biofilm is recognized as contributing substantially to increased resistance to antibiotics and innate host defense, due to structured-based limited diffusion or repulsion, which limits the efficacy of the antibiotics (Otto, 2006). However, we did not find a relationship between biofilm formation and antibiotic resistance, probably due to the small number of isolates used.

Despite the small number of clinical isolates, and not having identified the genes responsible for biofilm production (ica operon, polysaccharide intercellular adhesin) in order to correlate this gene with weak and strong production, important information regarding the antimicrobial resistance, as well as a new ST were determined. Nevertheless, future work is necessary to elucidate the role of S. epidermidis biofilm in antibiotic resistance.

In conclusion, S. epidermidis STs 2, 23, and 59 were present in two hospitals in different periods in Acapulco, Guerrero, Mexico. The clinical isolates obtained in this work showed a high multidrug resistance that is apparently not related to biofilm production. The duplex PCR standardized in this work, using specific primers for the nuc and mecA genes, is a reliable method to identify methicillin-resistant S. epidermidis that can be used to continue the epidemiological surveillance of HAIs. This study is the first report of S. epidermidis ST761.

Supplemental Information

Supplemental Information 1 Staphylococcus epidermidis data.

The results of antibiotic resistance (MICs), biofilm formation, and information on the clinical isolates used in this work.

Click here for additional data file.

Supplemental Information 2 Antibiotic MIC values for each S. epidermidis clinical isolate used in this study.

BP: benzylpenicillin, EM: erythromycin, CM: clindamycin, Q/D: quinupristin/dalfopristin, LE: levofloxacin, GM: gentamicin, TS: trimethoprim/sulfamethoxazole, RI: rifampicin, OX: oxacillin, LI: linezolid, VM: vancomycin, CX: cefoxitin, MIC: minimum inhibitory concentration, Interp.: interpretation, R: resistant, S: sensitive.

Click here for additional data file.

The authors would like to thank Ma. Elena Velazquez Meza (National Institute of Public Health) and Maria Román Salgado (Vicente Guerrero Hospital) for providing control strains and clinical isolates, respectively.

Additional Information and Declarations

Competing Interests

Author Contributions

Ethics

DNA Deposition

Data Availability

The authors declare that they have no competing interests.

Verónica I. Martínez-Santos conceived and designed the experiments, performed the experiments, analyzed the data, prepared figures and/or tables, authored or reviewed drafts of the article, and approved the final draft.

David A. Torres-Añorve performed the experiments, analyzed the data, prepared figures and/or tables, and approved the final draft.

Gabriela Echaniz-Aviles performed the experiments, analyzed the data, authored or reviewed drafts of the article, and approved the final draft.

Isela Parra-Rojas conceived and designed the experiments, authored or reviewed drafts of the article, and approved the final draft.

Arturo Ramirez-Peralta analyzed the data, authored or reviewed drafts of the article, and approved the final draft.

Natividad Castro-Alarcón conceived and designed the experiments, analyzed the data, prepared figures and/or tables, authored or reviewed drafts of the article, and approved the final draft.

The following information was supplied relating to ethical approvals (i.e., approving body and any reference numbers):

The University of Guerrero granted Ethical approval to carry out the study within its facilities (Ethical Application ref: CB-002/2021).

The following information was supplied regarding the deposition of DNA sequences:

The S. epidermidis new ST761 sequences are available at PubMLST: ST-761.

https://pubmlst.org/bigsdb?page=profileInfo&db=pubmlst_sepidermidis_seqdef&scheme_id=1&profile_id=761.

The following information was supplied regarding data availability:

The raw measurements are available in the Supplemental File.

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
