# Peer review of "Characterization of Staphylococcus epidermidis clinical isolates from hospitalized patients with bloodstream infection obtained in two time periods"

_PeerJ, doi:10.7717/peerj.14030_

## Round 0.1 · original submission · Major Revisions

Dear Dr. Martínez-Santos and Dr. Castro-Alarcón,

We have received reports from three independent referees. They all agreed with the relevance of the work in the research area concerning the antimicrobial resistance studies of S. epidermidis and the description of relevant strains and ST in Mexico. Likewise, the three reviewers indicated significant revisions.

The main criticisms are around -

i) standardized duplex PCR for identifying methicillin-resistant S. epidermidis, which needs to be explained in more detail,

ii) Regarding the experimental design, the small numbers of the isolates and biased sampling.

iii) The use of MLST and the duplex PCR methods, not the one proposed by Miragaia et al. (2008), to confirm the identification obtained with the automated system from Vitek®2.

Please review and respond to all points criticized by the referees in greater detail and time.

Sincerely
Marisa Fabiana Nicolás

Reviewer 1 ·

Basic reporting

The present study addresses an important issue in the management of infectious blood diseases, which is to characterize isolates of methicillin-resistant S. epidermidis obtained from blood cultures of patients from different services of two tertiary-level hospitals in Mexico. It confirms previous findings related to the characterization of this bacterium, and also it finds new results that it may be useful in the molecular epidemiology of the microorganism. They show that S. epidermidis has a high degree of genotypic diversity, as already found in other investigations. No association between the biofilm formation and resistance profiles were observed, as explained by the authors due to the small numbers of the isolates.
They also standardized a duplex PCR for the identification of methicillin-resistant S. epidermidis that, according to the authors, it could be useful tool since it is simple and easy, but not many details are given about the performance of the duplex-PCR.
However, the greatest limitation of the study is the small number of the isolates, and a very biased sampling, since only methicillin-resistant S. epidermidis isolates were studied. It is not clear how they were selected in order to assess their representativeness. The inclusion of isolates from two study periods without a clear epidemiological characterization of infections by this bacterium in the hospital or in the country is not well justified. The patients where the isolates came from were not characterized, it is only mentioned that it was one isolate per patient. It could be assumed that they are clinically significant isolates and not part of the patient's microflora or contamination during laboratory processing.
Figure legends are not included.
Figure 1 shows the bands of mecA and nuc, the band that corresponds to nuc is not very clear, it is as if dragged from the band of mecA.
The multi-resistance found in the study should not be surprising considering that the selected strains were already methicillin-resistant, which is associated with resistance to other ATB families.

Experimental design

Small numbers of the isolates and biased sampling.

Validity of the findings

Lack of representativity.

Reviewer 2 ·

Basic reporting

Overall, the text is clear and professional English was used. However, I would like to recommend minor modifications to improve the understanding. In line 67, I suggest changing the word mix to the word combination or a more formal synonym. In line 98, please review the spelling of bacteremia. In line 170, please check the number of antibiotics used. In lines 210/211, please consider replacing the use of the expression "of the latter" or rephrasing, the sentence is not easily clear. If the authors can afford it, I would recommend an English language editing service or a thorough review using some free available writing assistant. As antibiotic resistance is a trending topic and a world health concern, I expected to see more updated references, but it is not jeopardizing the context or the validity of the conclusions. The figures are 'raw' but informative, my only advice would be to improve the legend of Figure 2 because took me a couple of seconds to identify that the numbers were related to the STs identified in the work (without reading the cross-referenced text). I am not sure if the poor resolution is due to the pdf compilation, but please check the resolution of originally submitted figures (especially Figures 3 and 4). No further comments to add.

Experimental design

Although your experimental design is in accordance with the research question, some aspects of the methodology raised some issues:
1. How does the Vitek 2 automated system work? A brief description should be included.
2. What are the resistance breakpoints for each antibiotic tested? The authors based their classification (R or S) for all antibiotics tested on the CLSI guidelines? A brief description should be included.
3. A brief description of the goeBurst algorithm would help the reader to better understand what is the basis for the clonal relationship among the STs reported by the author.
4. What is the ground for the ODs cut-offs to classify the strains as biofilm non-, weak or strong producers?
5. As the biofilm experiments were made in triplicates and it is possible to see error bars in Figure 4 (some bigger than others), the authors applied some statistical methods to support the observed differences? Statistical analysis should be included before acceptance.

Validity of the findings

According to the authors, the aim of the work was to determine the most prevalent STs, as well as the antibiotic resistance profile and biofilm formation behavior of the collected S. epidermidis clinical isolates. In my opinion, the authors were able to achieve their goals. I commend the authors for pointing out the work limitations, as they recognize the small number of isolates addressed in the work without jeopardizing their conclusions and leaving a way open to future work. As stated above, statistical analysis should be included to support biofilm formation findings. Also, perhaps as supplemental material, the authors could add a table with the MIC for each antibiotic related to each isolate (is this information available when using this Vitek system?). In line 202, the authors stated they 'developed a duplex PCR', it is just a technicality, but I suggest rephase because the reader can be led to believe that the authors developed the multiplex PCR method, as the real intention is to state that they first applied the amplification of the two genes (nuc and mecA) together as a way to identify MRSE isolates. In addition, I suggest the authors, if it is possible, let the electrophoresis runs for a little while to better separate the DNA fragments (to avoid someone interpreting their results as DNA degradation, for instance). Other improvements could include PCR for each gene separately as control (using as a template the positive control strain) or yet working around the annealing temperature to improve/favor the amplification yield of the nuc gene. In line 166, the authors call the attention of the reader to some resistant isolation proportion comparison between the groups, but I do not see how this fact can be relevant, could be only a coincidence. Maybe consider rephrasing. My last observation is, in line 272, the authors raise a negative issue about not having identified the ica operon. I can not relate the context of this sentence with any other part of the text since this operon was not cited before. I suggest removing or better explaining: was it a technical impairment or did the authors not performed the analysis at all? If keeping, what is the relevance of this operon for the work hypothesis/conclusion?

Additional comments

No additional comments.

Reviewer 3 ·

Basic reporting

The Martínez-Santos et al. manuscript characterized clinical isolates of Staphylococcus epidermidis in two different periods (2003/2004 and 2017) from two hospitals in Acapulco, Guerrero according to ST, antimicrobial resistance profile, and biofilm formation. Overall, the manuscript is clearly written in unambiguous language.

Major Issues:

- Although the study adequately contextualizes the topics addressed, the last two paragraphs of the Introduction (lines 67 to 88) fail to relate to the subjects. The authors should rewrite the paragraphs to improve the understanding of the relationship between biofilm and antimicrobial resistance as well as the importance of molecular typing for the characterization of Staphylococcus epidermidis isolates, including the following published works:

1. Miragaia et al. “Comparison of Molecular Typing Methods for Characterization of Staphylococcus epidermidis: Proposal for Clone Definition”. Journal of Clinical Microbiology, 2008, Vol. 46, No. 1. https://doi.org/10.1128/JCM.01685-07

2. Fey & Olson. “Current concepts in biofilm formation of Staphylococcus epidermidis”. Future Microbiology, 2010, Vol. 5, No. 6. https://doi.org/10.2217/fmb.10.56

3. Foster. “Surface Proteins of Staphylococcus epidermidis”. Frontiers in Microbiology, 2020, Vol. 11. https://doi.org/10.3389/fmicb.2020.01829

- Line 36 of the Abstract has a different value from the Results section (lines 170-171) and Table 2. The correct value would be 17 or 85% of the isolates instead of 17%.

- The authors should write a paragraph in the Results section describing the new ST with all the characteristics identified in the manuscript, such as resistance profile, biofilm production, and molecular variability.

Minor Issues:

- Replace “bacteriemia” with “bacteremia” in lines 98 and 207

- A hyphen is missing in the: “methicillin resistant” (lines 28, 39, 89, 161, 252, and 280), “coagulase negative staphylococci” (line 49), and “multidrug resistant” (line 80).

- Correct the name of the species in line 40

- In lines 45 and 238 should be gram-positive

- Authors should improve the quality of Figure 1 because it is illegible

- The legend of Figure 2 is incorrect because the Venn Diagram shows the STs found in work and not the clinical isolates.

- In Figure 4, I suggest including the information regarding the ST

-The authors do not refer to the Supplementary Table in the text that should include information regarding the ST.

Experimental design

The methodology presented is well written and adequate for the results presented.

Major Issues:

- Miragaia et al. (2008) propose that clones within the S. epidermidis species be defined by the combination of the PFGE type followed by the SCCmec type. Please explain why you used the MLST and the duplex PCR methods and not the one proposed by Miragaia et al. (2008) to confirm the identification obtained with the automated system from Vitek®2.

- What was the identification rate of your method that makes it more accurate? How was the comparison made? The authors should specify in the text the advantages of the duplex PCR method and the reason they chose it for the identification of S. epidermidis.

- The "Use of the goeBurst algorithm" subsection needs to be more detailed, including the parameters used.

Minor Issues:

- The authors should include the references used to classify isolates as multidrug-resistant (line 171) and as biofilm producers (lines 156-157).

Validity of the findings

The methodology applied is robust enough to support the conclusions of the manuscript, except for the choice of the standardized duplex PCR method that needs to be better contextualized throughout the text. Nevertheless, the article provides relevant information about the distribution and phenotypic characteristics of the strains circulating in Mexico in distinct periods, adding knowledge about the antibiotics that would be most effective in infections in the region studied. In addition, the manuscript identifies and provides the characteristics of a new ST, essential for understanding the worldwide dispersion of strains of clinical interest. Therefore, I endorse the publication of the manuscript after the suggested corrections.

---

## Round 0.2 · accepted · Accept

The reviewers have not suggested any new corrections and agree with the manuscript's publication.

Reviewer 1 ·

Basic reporting

No comments

Experimental design

No comments

Validity of the findings

No comments

Additional comments

I appreciate the effort of the authors for understanding and accepting my comments on the previous reviews.
In my opinion, the article is ready for publishing.

Reviewer 3 ·

Basic reporting

No comment

Experimental design

No comment

Validity of the findings

No comment

Additional comments

The authors corrected the manuscript according to the suggestions, and it is suitable for publication.